# Using Machine Learning Methods to Provision Virtual Sensors in Sensor-Cloud

**DOI:** 10.3390/s20071836

**Published:** 2020-03-26

**Authors:** Ming-Zheng Zhang, Liang-Min Wang, Shu-Ming Xiong

**Affiliations:** School of Computer Science and Communication Engineering, Jiangsu University, Zhenjiang 212013, China; 2211708015@stmail.ujs.edu.cn (M.-Z.Z.); wanglm@ujs.edu.cn (L.-M.W.)

**Keywords:** sensor-cloud, agricultural IoT, virtual sensor provisioning, machine learning, representative sensors

## Abstract

The advent of sensor-cloud technology alleviates the limitations of traditional wireless sensor networks (WSNs) in terms of energy, storage, and computing, which has tremendous potential in various agricultural internet of things (IoT) applications. In the sensor-cloud environment, virtual sensor provisioning is an essential task. It chooses physical sensors to create virtual sensors in response to the users’ requests. Considering the capricious meteorological environment of the outdoors, this paper presents an measurements similarity-based virtual-sensor provisioning scheme by taking advantage of machine learning in data analysis. First, to distinguish the changing trends, we classified all the physical sensors into several categories using historical data. Then, the *k*-means clustering algorithm was exploited for each class to cluster the physical sensors with high similarity. Finally, one representative physical sensor from each cluster was selected to create the corresponding virtual sensors. The experimental results show the reformation of our scheme with respect to energy efficiency, network lifetime, and data accuracy compared with the benchmark schemes.

## 1. Introduction

In recent years, motivated by the incorporation of the ubiquitous environmental information sensing ability of WSNs as well as the powerful data storage and processing capabilities of cloud computing, sensor-cloud technology has been receiving growing attention in various application domains [1], such as target tracking [2], environment monitoring [3], smart city [4], public safety system [5], and precision irrigation [6]. It conveys a concept of providing physical sensors as a service (Se-aaS) [7]. In the sensor-cloud framework, various physical sensors and the corresponding cloud services are provided by sensor owners and cloud service providers, respectively. The end-users can get the information from these infrastructures without actually owning them. They only need to send their service requests to the sensor-cloud platform and pay for usage. Considering that the end-users are usually low-income farmers in agriculture fields, this pay-per-use model may alleviate the burden of farmers from the high cost and onerous farmland tasks [8]. Furthermore, the sensor-cloud framework enables multiple users to share the same infrastructure by using virtualization technology. Thus, the profits of the service providers can also be guaranteed.

The sensor-cloud framework is viewed as a paradigm shift from traditional WSNs. It decouples the physical sensor (data producer) from the data provider, which is conceptualized as a virtual sensor working in the virtual machine [9]. In some early works, researchers have focused on the design of system architecture. For instance, the authors in [9] described the design and the operation mode of the sensor-cloud framework. In [10], the authors gave a basic definition and interrelation of each component in sensor-cloud mathematically. Similarly, the authors in [11] introduced the sensor-cloud infrastructure based on IoT-cloud. They divided the whole system framework into three layers: the client layer, the middleware layer and the physical layer. The client layer provides various interfaces to users. Thus, the users can access the website and request the services provided by the sensor-cloud platform. When the user’s requests arrive, the middleware layer will allocate suitable physical sensors to create a virtual sensor in response to the user’s requests.

The existing studies on virtual sensor provisioning have mainly focused on data aggregation or spatial correlation. In [12], all the physical sensors were activated to collect environmental data. Then, the collected sensing data were processed in the cloud. However, this method results in increased energy consumption. Hence, the underlying network’s lifetime will be reduced. Thereby, the sensor owners must redeploy their invalid physical sensors to maintain the services. As a result, the cost for users also increases. The authors in [13] gave the same inference. Thereafter, the authors in [14] proposed a cluster-based virtual sensor provisioning method. In this paper, the physical sensors with spatial and measurement similarities were clustered into one cluster, then, the ant algorithm was exploited to select physical sensors to provision virtual sensors. In this way, numerous physical sensors were switched to the dormant state, avoiding more extra energy consumption. However, this method only considers the feedback of the current sensing data. The historical data stored in the cloud server were fully not considered.

According to the existing studies, machine learning methods are widely used for data analysis in various industries. For instance, vehicular social networks [15], cyber security [16], and smart agriculture [17]. Leveraging the powerful storage and computing capability of cloud computing, the massive sensing data can be stored and processed efficiently. As we all known, the data collected from the WSNs have a certain spatial-temporal similarity [18]. Motivated by this, when implementing the virtual sensor provisioning task, the physical sensors with high similarity are clustered into one cluster. Then, several representative physical sensors were chosen from each cluster to provide data collection services. The residual physical sensors are switched to the dormant state, which reduces the overall energy consumption and prolongs the network lifetime. Considering that, the physical sensors with a close distance may have different measurements. For instance, if there are some trees in the cropland, the temperature or light around the trees may be lower than in other places. Thus, in this paper, we focused on the similarity of sensing data and exploit machine learning methods to optimize the selection of physical sensors.

Classification and clustering are commonly used data analysis methods in machine learning. For a given sample set, also named a dataset, the goal of a clustering algorithm is to segment the whole dataset into several clusters (i.e., subsets). Thus, samples within the same clusters are more similar to each other than those in different clusters. In this paper, we exploited the *k*-means clustering algorithm to cluster physical sensors according to their sensing data. Moreover, considering that, the physical sensors with the similar measurement values may have different changing trends. For instance, there are two temperature sensors ($Sensor_{A}$ and $Sensor_{B}$), both of which measurement values are 25 °C. However, the temperature value of $Sensor_{A}$ is rising, and the value of $Sensor_{B}$ is falling. In this condition, we think that $Sensor_{A}$ and $Sensor_{B}$ should be divided into two different clusters. Thus, before implementing the clustering algorithm, the linear regression method is applied to analyze each physical sensor’s changing trend according to the historical data. The main contributions of this article are outlined as follows:(1)We propose an energy-efficient virtual sensor provisioning scheme based on the similarity of sensing data. Differently from the spatial similarity based scheme, the physical sensors in our scheme with the highest correlation of measurement values can be divided into one cluster, even though they are far from each other in geographical areas.(2)To ensure the accuracy of results, we first use the linear regression model to classify all the physical sensors into several classes according to the historical data, then exploit the *k*-means clustering algorithm to optimize the selection. As a result, we use fewer physical sensors to provide a higher quality of service.(3)In addition to paying attention to the changing trends of physical sensors, we also consider the number of sensory parameters. The sensors chose in our schemes can sense two kind parameters that differ from the scheme using single parameter sensors.

The rest of this article is organized as follows: Section 2 introduces some related works and a brief discussion. Section 3 describes the system model and the problem definition of virtual sensor provisioning. In Section 4, we detail the proposed machine learning based virtual sensor provisioning scheme in detail. Then, Section 5 evaluates the performance of our proposed scheme and analyzes the simulation results. Finally, the conclusions and future works are drawn in Section 6.

## 2. Related Works

Virtual sensor provisioning is an important task in sensor-cloud, which is similar to the sensor allocation in traditional WSNs. In contrast, exploiting the virtualization technology, the physical sensors in the sensor-cloud framework can be shared with multiple users via the virtual sensors. The main goal of virtual sensor provisioning is to reduce energy consumption and redundancy data. With the same goal in traditional WSNs, the low energy adaptive clustering hierarchy (LEACH) protocol is applied widely [19,20]. In this method, the cluster head nodes are selected randomly with probability *p*, and the data packages from cluster members are sent to the sink node via the cluster head nodes. The cluster head nodes are selected periodically so that the energy load of the entire network can be allocated evenly to each sensor node. However, this method does not consider the multiple user’s request specifically, so it is not suitable in the sensor-cloud environment.

The authors in [21] designed an agricultural sensor-cloud framework to provide multiple services for farmers. In order to ensure that the event data packets can be forwarded to the sink fast and reliably, the authors focused on the routing protocol of the physical layer and presented a priority-based data transmission technique. However, they tended to activate all the deployed physical sensors for virtual sensor provisioning. Although this method can utilize the powerful ability of cloud computing to process the massive data packages efficiently, the lifetime of WSNs will reduce with the increase of energy consumption. Similarly, the authors in [22] proposed a software-defined network (SDN) based load balancing and low response delay edge-cloud network framework, which reduces the redundant data and service response time.

In [12], the authors divided the whole geographical area into several regions. The data from the same region were sent to the middleware layer and processed with a hierarchical data aggregation method. The authors introduced four different virtual sensor configurations, such as one-to-many, many-to-one, many-to-many, and derived, which are the basic forms of physical sensor virtualization. However, the network-level virtualization [23] has not fully utilized the advantages of sensor-cloud infrastructure. Furthermore, the physical sensors from the same region do not guarantee the data correlation to perform the aggregation. As mentioned above, if there are some trees in the cropland, the temperature and humidity in the woods may be quite different from the outside.

Thereafter, some researchers take into account the spatial-temporal correlation when creating virtual sensors. They consider the dynamic correlation of the sensing data and propose an active node selection algorithm to reduce energy consumption [24]. The values of dormant sensors can be predicted by the activated sensors with high spatial-temporal correlation. An integration model based on historical data prediction is proposed in [25]. The authors exploited the spatial-temporal correlation to predict and controlled the accuracy of the sensing data, which provides a trade-off between the quality of service and energy efficiency.

The authors in [26] mainly focused on the similarity of sensing data. When the user initiates a service request, all the physical sensors are activated, then the collected data are sent to the middleware layer via the sink nodes for further processing. The middleware clusters the physical sensors based on the similarity of current sensing data. The generation of clusters is regulated by the predefined mean squared error. In this way, the physical sensors in the same cluster may be distributed in different areas. However, this scheme does not fully consider the changing trends of the environment (e.g., example of $Sensor_{A}$ and $Sensor_{B}$ in Section 2). In this case, direct data aggregation may lead to inaccurate results. Thus, before clustering the physical sensors, it is necessary to perform a classification process in advance.

As for the node selection algorithm, the authors in [27] proposed an adaptive clustering algorithm in a multihop mobile network. This algorithm partitions the whole network into several disjoint clusters according to each node’s 1-hop neighbors. Similarly, in [28], a coalition-head selection algorithm is present to support the trapped users to form a coalition based on users’ transmission range. These two node selection algorithms are similar to the allocation scheme based on spatial correlation as mentioned above. Thus, it is difficult to distinguish the physical sensors with different changing trends. Furthermore, these algorithms are focused on mobile networks. The network topology is unstable. In our scheme, we mainly focused on the static network.

From the above discussion, the existing works about virtual sensor provisioning are mainly focused on activating all physical sensors, network-level virtualization, spatial correlation, or data analysis for current sensing data. These methods may result in more energy consumption and redundant data. In addition, the role of historical data has not been fully considered.

## 3. System Model

Figure 1 shows the basic architecture of agricultural sensor-cloud, which can be divided into three layers: the user layer, the middleware layer, and the physical layer [7].

(1)The user layer: The user layer is responsible for providing standard interfaces and various application services, which can be applied for users with different operating systems (e.g., Windows, Linux or Android). They are mainly two kinds of users: farmers and device owners. Farmers can request all the applications according to their demands. Device owners are the providers of the infrastructural, they should provide all the information about the sensors when registering on the platform.(2)The middleware layer: The middleware layer acts as an intermediary between the user layer and the physical layer, which is responsible for agricultural data analysis, storage, and virtual sensor provisioning. Leveraging the benefits of cloud computing, the sensing data can be processed efficiently. Users can obtain these data from the corresponding services. Each service may consist of multiple virtual sensors, and each virtual sensor may belong to multiple services.(3)The physical layer: The physical layer contains various physical sensors and agricultural devices, which are applied for data communication and implementing the users’ strategies, respectively. In this layer, each physical sensor transmits its information (e.g., position, type, and values) to the corresponding sink through the ZigBee network, which is assumed to be star topology here for simplicity description [8]. Then, the sink will act as relay nodes and forward the collected data to the cloud platform using 4G [29] or 5G [30] networks.

We use an example to depict the working pattern. Assume that a farmer wants to request an irrigation service. Thus, he (she) needs to know the information about soil moisture. The request is submitted to the sensor-cloud platform. As the request arrives, the middleware will send a query to activate all the related physical sensors. Then, each physical sensor sends its sensing data (i.e., soil moisture) to the cloud via the corresponding sink (i.e., sink_1). Thereafter, virtual sensors provisioning task will be implemented to choose the most representative physical sensors in response to the farmer’s request. These physical sensors will be active until the service time is over and the other physical sensors will switch to dormant to reduce energy consumption. Finally, the farmer will make an irrigation decision based on the received sensing data.

### Problem Definition

In this section, we introduce the basic components of the sensor-cloud framework and present the definition of virtual sensor provisioning mathematically. Figure 2 shows the virtualization model.

(1)The physical sensor: A physical sensor (*P*) is the basic resource unit to create virtual sensors. They are deployed by sensor owners to monitor the specific environment parameters. The sensor owners should provide the information (e.g., identifier, position, and type) of each physical sensor when registering in the portal of sensor-cloud platform. Moreover, the position of physical sensors is usually fixed. For each physical sensor ($P_{i}\in P$), it can be defined as a tuple with the following parameters:
(1)$$P_{i}=\left\{P_{id},P_{loc},P_{type},P_{m},P_{\Psi }\right\},$$
where id, loc, type, *m*, and Ψ denote the physical sensor’s identifier, location, parameter type (e.g., temperature or humidity), measurement value, and state (e.g., active or dormant), respectively. Moreover, the set of measurement values, also named sample set or dataset, can be defined as a matrix *X*:
(2)$$X=\left[\begin{array}{cccc} x_{11} & x_{12} & \cdots & x_{1n}\\ x_{21} & x_{22} & \cdots & x_{2n}\\ \vdots & \vdots & \; & \vdots \\ x_{m1} & x_{m2} & \cdots & x_{mn} \end{array}\right],$$
where the *i*th row indicates the *i*th attribute, i=1,2,…,m; the *j*th column indicates the *j*th sample, and j=1,2,…,n. For instance, assuming that the used physical sensor can sense three parameters: 1-temperature, 2-humidity, and 3-light. Thus, $a_{13}=25$ means the measurement value of the first attribute of sensor node 3 (i.e., temperature) is 25 °C.

(2)The virtual sensor: The virtual sensor (*V*) can be regarded as a data provider in sensor-cloud and obtains its data from the underlying physical sensors. It can have a data processing program to process the sensing data in response to the user’s service requests. Similarly, for each $V_{i}\in V$, it can be defined as:
(3)$$V_{i}=\left\{V_{id},V_{type},V_{p},V_{\Psi }\right\},$$
where the id, type, *p*, and Ψ denote the virtual sensor’s identifier, service type, used physical sensors, and state (e.g., active or terminated), respectively. Generally, it is not necessary to activate all the physical sensors to provision virtual sensors. From our point of view, each virtual sensor can be viewed as a logical mapping of multiple physical sensors with the same or different types. The mapping function between $P_{i}$ and $V_{i}$ can be expressed as:
(4)$$V_{i}=f_{\xi }\left(P_{i}\right),V_{i}\in V,P_{i}\in P,$$
where ξ is the mapping relation between $P_{i}$ and $V_{i}$, such as one-to-many, many-to-one, many-to-many, or derived.

(3)Service: The sensor-cloud platform provides various services (*S*) (e.g., irrigation, weather, or GPS service) for users. They can log in to the website via the browser or mobile device and choose the services according to their demands. A service ($S_{i}$) may be supported by multiple virtual sensors. For instance, an irrigation service may contain the following virtual sensor services: soil moisture, air humidity, and temperature. Here, $S_{i}$ can be denoted as:
(5)$$S_{i}=\left\{S_{own},S_{type},S_{area},S_{\tau },S_{t}\right\},$$
where own, type, area, τ, and *t* denote the service’s requester, type, coverage area, data collection interval (e.g., once-per-30s or other), and total service time, respectively.

(4)Virtual sensor provisioning: After receiving the user’s service request, the middleware will retrieve the catalog of physical sensors in the service area. Then, all the related physical sensors send their sensing data to the sensor-cloud platform via the sink node. The middleware will combine these fresh data with the historical data stored in the storage server, grouping these physical sensors into several clusters according to the measurement similarity. Thus, for $x_{i},x_{j}\in X$, $x_{i}={\left\{x_{1i},x_{2i},\ldots ,x_{mi}\right\}}^{T}$, $x_{j}={\left\{x_{1j},x_{2j},\ldots ,x_{mj}\right\}}^{T}$, if $x_{i}$ and $x_{j}$ belong to the same cluster, then there is:
(6)$$d_{ij}={\left(\sum_{\phi =1}^{m}{\left|x_{\phi i}-x_{\phi j}\right|}^{p}\right)}^{\frac{1}{p}}\leq D,$$
where $d_{ij}$ is the distance or similarity of the $x_{i}$ and $x_{j}$, *D* is the given threshold of distance.For each cluster, only one representative physical sensor $P_{r}$ is selected to activate. The residual physical sensors will switch to dormant to save energy. As a result, all the representative nodes create a virtual sensor in response to the user’s request. This process is called virtual sensor provisioning.**Problem definition:** For a given substrate physical sensor set (*P*). The task of virtual sensor provisioning is to find the optimal subset $P^{\prime }$ to create a virtual sensor ($V_{i}$) in response to the user’s request. In this paper, we implement the virtual sensor provisioning as a clustering problem. The object is to select the representative physical sensor from each cluster. Finally, achieving a decrease in energy consumption and redundant data, meanwhile, improving the service quality.

## 4. Proposed Virtual Sensors Provisioning Scheme

In our scheme, we exploited the *k*-means algorithm to implement the virtual sensor provisioning. Moreover, considering that the physical sensors with similar values may have different changing trends, the linear regression model was applied to distinguish the changing trends.

### 4.1. Model

Assume that there are *n* physical sensors deployed in the monitoring area. Their sensing data compose a sample set $X=\{x_{1},x_{2},\ldots ,x_{n}\}$, which can be denoted by an eigenvector with *m* dimensions, such as time ($m_{1}=t$), node identifier ($m_{2}=id$), and sensing parameter ($m_{3}=temperature$). The first step is to classify *n* samples into *l* groups (*G*) according to the changing trends, in which:(7)$$\begin{array}{c} G=\{G_{1},G_{2},\ldots ,G_{l}\},l<n,\;\mathrm{and}\\ G_{i}\cap G_{j}=\emptyset ,\left(i\neq j\right),\;\;\overset{l}{\underset{i=1}{⋃}}G_{i}=X. \end{array}$$

Then, clustering each group $\left(G_{i}\in G\right)$ into $k_{i}$ different clusters ($C_{i}$):(8)$$\begin{array}{c} C_{i}=\left\{C_{i}^{1},C_{i}^{2},\ldots ,C_{i}^{k_{i}}\right\},\;\mathrm{and}\\ C_{i}^{k_{i}}\cap C_{i}^{k_{j}}=\emptyset ,\left(k_{i}\neq k_{j}\right),\;\;\overset{k_{i}}{\underset{1}{⋃}}C_{i}=G_{i}. \end{array}$$

From the final results, *n* samples are clustered into *k* clusters. So, the relationship between *n* and *k* can be expressed as: γ=C(i), in which i∈{1,2,…,n}, γ∈{1,2,…,k}. The model of *k*-means clustering is a many-to-one function from samples to clusters.

### 4.2. Strategy

The linear regression model is applied to classify these physical sensors. For a sample $x=\{x_{1},x_{2},\ldots x_{m}\}$, there is:(9)$$f\left(x_{i}\right)=\omega x_{i}+b,$$
the object of linear regression is to get the optimal value of ω and *b* to achieve $f\left(x_{i}\right)≃y_{i}$. Here, $y_{i}$ is the actual value. The value of ω can be calculated by:(10)$$\omega =\frac{\sum_{i=1}^{m}y_{i}\left(x_{i}-\bar{x}\right)}{\sum_{i=1}^{m}x_{i}^{2}-\frac{1}{m}{\left(\sum_{i=1}^{m}x_{i}\right)}^{2}},$$
where $\bar{x}=\frac{1}{m}\sum_{i=1}^{m}x_{i}$ is the average of *x*. We also define a flag (θ) to denote the changing trend of each attribute (*j*), it can be expressed as:(11)$$\theta_{j}=\left\{\begin{array}{cc} 0 & \omega_{j}\leq 0\\ 1 & \omega_{j}>0 \end{array}\right..$$

Thus, for each sample x∈X, the overall changing trend can be denoted by a flag set $\Theta_{i}=\left\{\theta_{1},\theta_{2},\ldots ,\theta_{m}\right\}$.

After the initial classification, we apply the *k*-means clustering algorithm to select the optimal division ($C^{*}$) for each group by minimizing the Loss Function (L). In this paper, we adopt *Squared Euclidean Distance (SED)* as the index to evaluate the distance $d(x_{i},x_{j})$ or similarity of samples.
(12)$$\begin{array}{cc} d(x_{i},x_{j}) & =\sum_{\rho =1}^{m}{\left(x_{\rho i}-x_{\rho j}\right)}^{2}\\ \; & =\|x_{i}-x_{j}{\|}^{2}, \end{array}$$
where ρ denotes the ρth dimension of sample. L is the sum of the distances between each sample and the center of the cluster to which it belongs, it denotes the similarity of the samples in the same cluster.
(13)$$L=\sum_{\gamma =1}^{k}\sum_{C(i)=l}{\left\|x_{i}-{\bar{x}}_{\gamma }\right\|}^{2},$$
where ${\bar{x}}_{\gamma }={\left({\bar{x}}_{1\gamma },{\bar{x}}_{2\gamma },\ldots {\bar{x}}_{m\gamma }\right)}^{T}$ is the center of the γth cluster. Thus, the *k*-means clustering can be viewed as an optimization problem with the object:(14)$$\begin{array}{cc} C^{*} & =\arg\min_{C}L\\ \; & =\arg\min_{C}\sum_{\gamma =1}^{k}\sum_{C(i)=\gamma }{\left\|x_{i}-x_{j}\right\|}^{2}. \end{array}$$

The value of $C^{*}$ reaches a minimum when similar samples are gathered into the same cluster. The task of dividing *n* samples into *k* clusters is a combined optimization problem; thus, the number of all possible clustering results can be calculated by:(15)$$N\left(n,k\right)=\frac{1}{k!}\sum_{\gamma =1}^{k}{\left(-1\right)}^{k-\gamma }\left(\begin{array}{c} k\\ \gamma \end{array}\right)k^{n}.$$

From the Equation (Equation 15), we can know that the result of N is exponential because the optimal solution of the *k*-means clustering problem is *NP-hard*. The iterative method is commonly applied to solve this problem.

### 4.3. Algorithm

Algorithm 1 details the entire virtual sensor provisioning scheme, which can be divided into two steps: classification and clustering. Initially, there is only one physical sensor in each group (i.e., $G_{i}=\left\{i\right\},1\leq i\leq n$). Subsequently, the linear regression model is applied to calculate the value of $\Theta (G_{i})$ according to the historical data. The physical sensors with the same value of Θ are grouped into one class. Then, in clustering, for each class, the first step is to assign each sample to its nearest centroid (i.e., $\mu \left(G_{i}\right)=\left\{\mu_{1},\mu_{2},\ldots ,\mu_{k_{i}}\right\}$), and the second step is to create new centroids by taking the mean value of all the samples assigned to each previous centroid. The difference between the old and the new centroids is computed and the algorithm repeats these last two steps until this value is less than a threshold. In other words, it iterates until the centroids do not move significantly.
**Algorithm 1** Virtual sensor provisioning algorithm.**Input:** Sample set *X*
 1:**Step1:Classification** 2:Calculate the value of ω for each attribute. 3:$G=\{G_{1},G_{2},\ldots ,G_{n}\}$ 4:**for**1≤i<j≤n**do**   5: **if**
$\Theta \left(G_{i}\right)=\Theta \left(G_{j}\right)$
**then** 6:  $G_{i}=G_{i}\cup G_{j}$ 7: **end if** 8:**end for** 9:**Return**$G=\{G_{1},G_{2},\ldots ,G_{l}\}$10:**Step2:Clustering**11:$C_{j}=\emptyset ,\left(1\leq j\leq k_{i}\right)$12:$\mu \left(G_{i}\right)=\left\{\mu_{1},\mu_{2},\ldots ,\mu_{k_{i}}\right\}$13:$\lambda_{i}=\arg\min d_{ij}$14:$C_{\lambda_{i}}=C_{\lambda_{i}}⋃\left\{x_{i}\right\}$15:**for**$j=1,j++,j\leq k_{i}$**do**16:$\mu {\left(G_{i}\right)}^{{}^{\prime }}=\frac{1}{|C_{j}|}\sum_{x_{i}\in C_{j}}x_{i}$, **then**17: **if**
$\mu \left(G_{i}\right)\neq \mu {\left(G_{i}\right)}^{{}^{\prime }}$18:  Update $\mu \left(G_{i}\right)=\mu {\left(G_{i}\right)}^{{}^{\prime }}$19: **end if**20:**end for**
**Output:**
$C^{*}=\left\{C_{1}^{*},C_{2}^{*},\ldots ,C_{l}^{*}\right\}$

### 4.4. Computational Complexity

For the given dataset $X_{mn}$. The computational complexity can be divided into two phase according to the algorithm’s execution steps. In classification phase, the time complexity is $O(m^{2}\left(m+n\right))$, and the space complexity is O(mn); In clustering phase, the time complexity is O(tkmn), and the space complexity is O(m(n+k)). Here, *t* is the number of iterations. The proof is given in the following.

**Proof** **1.**We use the least squares method to train the model. For a sample $x_{j}=\left(x_{1},x_{2},\ldots ,x_{m}\right),j=1,2,\ldots ,n$. The linear regression model can be defined as:
(16)$$h_{\xi }=\xi_{0}+\xi_{1}x_{1}+\ldots +\xi_{m}x_{m},$$
in which $\xi =(\xi_{0},\xi_{1},\ldots ,\xi_{m})$. For simplify, we set $x_{j}=\left(x_{0},x_{1},\ldots ,x_{m}\right),x_{0}=1$. Then Equation (Equation 16) can be expressed as:
(17)$$h_{\xi }=\xi^{T}x,$$
the mean square error is applied as the cost function:
(18)$$J\left(\xi_{0},\xi_{1},\ldots ,\xi_{n}\right)=\frac{1}{2n}\sum_{i=1}^{n}{\left(h_{\xi }\left(x^{\left(i\right)}\right)-y^{\left(i\right)}\right)}^{2},$$
where $x^{\left(i\right)}$ denotes the feature vector of the *i*th sample, and $y^{\left(i\right)}$ denotes the actual value of the *i*th sample, *n* is the number of samples. Thus, $X=\left(x^{\left(1\right)},x^{\left(2\right)},\ldots ,x^{\left(n\right)}\right)\in R^{n(m+1)}$, and $Y=\left(y^{\left(1\right)},y^{\left(2\right)},\ldots ,y^{\left(n\right)}\right)\in R^{n}$. So, the cost function can be expressed as:
(19)$$J\left(\xi \right)={\left(Y-X\xi \right)}^{T}\left(Y-X\xi \right),$$
calculate the derivative of ξ:
(20)$$\frac{\partial J(\xi )}{\partial \xi }=2X^{T}\left(X\xi -Y\right),$$Set $\frac{\partial J(\xi )}{\partial \xi }=0$, we can get:
(21)$$\xi ={\left(X^{T}X\right)}^{-1}X^{T}y,$$
the most computationally intensive part is $X^{T}X$, the time complexity is $O(nm^{2})$, and then inverting it, the time complexity is $O(m^{3})$. Thus, the overall time complexity is $O(m^{2}\left(m+n\right))$. Furthermore, it only need to store the dataset $X_{mn}$, so the space complexity is O(mn). □

**Proof** **2.**The implementation of *k*-means clustering algorithm for each class can be divided into four steps:
(1)Select *k* samples as the initial centroids, $\mu =\{\mu_{1},\mu_{2},\ldots ,\mu_{k}\}$;(2)Calculate the distance between sample $x_{j}$ and μ. Then divide $x_{j}$ into the cluster corresponding to the centroid with the smallest distance;(3)For each new cluster ($C_{i}$), calculate the new centroid $\mu^{\prime }=\frac{1}{|C_{i}|}\sum_{x_{j}\in C_{i}}x_{j}$;(4)Repeat the above two steps until the centroids do not move significantly.□

In sum, the time complexity in step 2 is O(km), in which *k* is the number of clusters, and *m* is the dimension of a sample. However, there are *n* samples and *t* iterations. Thus, the overall time complexity is O(tkmn). In addition, it only need to store the samples and centroids, so the space complexity is O(m(n+k)).

## 5. Performance Evaluation

### 5.1. Simulation Settings

The experiment of virtual sensor provisioning is done using Python3 language (https://www.python.org/) and NS-3 (https://www.nsnam.org/). Python3 is responsible for processing the dataset and implementing the clustering algorithm. In this paper, the dataset used comes from Intel Lab Data (http://db.csail.mit.edu/labdata/labdata.html). This dataset consists of 54 physical sensors that collect the environmental data once every 31 s. But there are massive null values and outliers, the record number of each node is also different. Some nodes only have a little record. To guarantee the accuracy of results, we selected 50 sensors with sufficient data. The used parameters are temperature and humidity. As for NS-3 (version:3.28), it is applied to simulate the performance of the generated underlying network. In the simulation, the location of each physical sensor is random, so we adopt the average of three simulation results. The simulation area is 100 m × 100 m, assuming that each node can communicate with the sink node directly. The initial energy of each node is 0.3 j. Table 1 lists all the used simulation parameters.

### 5.2. Performance Metrics

In this section, we analyze the performance of our proposed scheme in terms of energy consumption, network lifetime, and data accuracy. We briefly present the corresponding definitions in the performance evaluation.

(1)Energy consumption: We exploit the average energy consumption of the whole network to evaluate the energy efficiency of our scheme and the benchmark schemes. In each round, the energy consumption of our sensor-cloud based scheme and the LEACH scheme can be calculated by:
(22)$$E_{sc}=\sum_{k=1}^{k}\left(E_{k}^{s}+E_{k}^{t}\right),$$
where *k* is the number of selected physical sensors, $E_{k}^{s}$ and $E_{k}^{t}$ is the energy consumption due to sensing and transmission.
(23)$$E_{leach}=\sum_{p=1}^{n}\left(E_{p}^{s}+E_{p}^{t}+E_{p}^{r}+E_{p}^{c}\right),$$
where *n* is the number of physical sensors, $E_{p}^{s}$, $E_{p}^{t}$, $E_{p}^{r}$, and $E_{p}^{c}$ are the energy consumption due to transmission, receiving, sensing, and computing, respectively.(2)Network lifetime: We define the network lifetime as in the percentage of network which has not yet depleted their energy or their energy level is greater than the predefined threshold. In this paper, we choose the number of iteration round when the total remaining energy is above 10% as the evaluation indicator.(3)Data accuracy: The proposed scheme groups all the physical sensors into several clusters and chooses some representative nodes to provide services. However, whether these selected physical sensors can represent the whole network is uncertain. Thus for any cluster ($C_{i}^{*}$), we exploit Equation (Equation 24) to evaluate the data accuracy of our scheme.
(24)$$\delta_{i}=\sqrt{\frac{1}{N}\sum_{j=1}^{k}{\left(M_{r}-M_{j}\right)}^{2}},\left(1\leq j\leq l\right),$$
where δ is the evaluation indicator of data discrete degree, $\delta_{i}$ is used to calculate the data accuracy of our scheme, respectively. *N* is the number of records, *k* is the number of physical sensors in cluster $C_{i}^{*}$ except the representative node, $M_{r}$ is the measurement value of representative node, and $M_{j}$ is the measurement value of the residual node *j*.Similarly, Equation (Equation 25) are used to evaluate the data aggregation scheme in [12].
(25)$$\delta_{i}^{\prime }=\sqrt{\frac{1}{N}\sum_{j=1}^{k}{\left(\bar{M}-M_{j}\right)}^{2}},\left(1\leq j\leq l\right),$$
where $\delta_{i}^{\prime }$ is used to calculate the data accuracy of data aggregation scheme, and $\bar{M}$ is the average of all the physical sensors. In addition, the overall data accuracy (Δ) can be obtained by calculating the average of δ.
(26)$$\Delta =\frac{1}{l}\sum_{i=1}^{l}\delta_{i}.$$

### 5.3. Benchmark

We compare our scheme with the LEACH scheme in traditional WSNs and the data aggregation scheme in [12]. In the LEACH protocol, the network selects cluster head nodes randomly with probability *p*. The cluster member nodes send the data packages to the cluster head nodes. The cluster head nodes aggregate and transmit these data packages to the sink node. After a period, the network reselects new cluster head nodes based on the remaining energy. Moreover, the data aggregation scheme utilizes all the physical sensors to provide data collection services. The virtual sensors are created based on the geographical position of physical sensors, and their values are the average of the physical sensors.

### 5.4. Results and Analysis

The classification task is completed by a Python3 linear regression script. Since the data of light in the used dataset is incomplete, so only temperature and humidity parameters are considered in our experiments. As a result, 50 sensor nodes are divided into 4 classes according to their changing trends. The classification results are shown in Table 2.

Then implementing the *k*-means clustering algorithm for each class, the first is exploiting the *Elbow Method (EM)* to select a suitable number of clusters ($k_{i}$). The core index of EM is the sum of squared errors, its value will decrease as $k_{i}$ increases. We set $k_{1}=1$ because of there are only 3 nodes, and Figure 3 shows the relationship between $k_{i}$ and SSE. As a result, k={1,3,4,3}.

Our proposed scheme divides all the 50 physical sensors into 11 clusters. Then in each cluster, only one physical sensor is selected to provision the virtual sensor. Table 3 shows the clustering results.

Figure 4 analyzes the performance of our scheme and data aggregation scheme in [12] with respect to data accuracy. For simplicity, we take the class (0, 1) as an example. From Figure 4a–c, our scheme can cluster the physical sensors with high similarity to one set accurately. Although the values of Figure 4a,b are similar to each other, the temperature values of nodes 10 and 16 are higher than 14 and 15, they are divided into two different clusters. Thus, we can choose nodes 10, 14, and 19 to create a virtual sensor and provide service. As for the data aggregation scheme, it exploits all 6 physical sensors in class (0, 1). The data provided to users are the average of these physical sensors. As shown in Figure 4d.

In this paper, we select 300 consecutive data to estimate the data accuracy. In our scheme (labeled as ‘os’), the value of δ can be calculated according to Equation (Equation 18): $\delta_{10}^{T}=0.8797$, $\delta_{10}^{H}=1.2364$; $\delta_{14}^{T}=0.6376$, $\delta_{14}^{H}=1.6984$; $\delta_{19}^{T}=1.0537$, $\delta_{19}^{H}=1.0439$; $\Delta_{os}=\left\{T:0.8570,H:1.3262\right\}$. The results are shown in Table 4.

Similarly, the data accuracy of the data aggregation scheme can be obtained by Equation (Equation 19): $\delta_{agg}^{T}=\left\{1.3297,0.4875,2.2456,1.6793,3.3555,2.3367\right\}$, $\delta_{agg}^{H}=\left\{3.5117,4.5019,1.8327,3.3690,0.9170,2.0327\right\}$; $\Delta_{agg}=\left\{T:1.9089,H:2.6942\right\}$. The results are shown in Table 5.

In sum, our scheme achieves 55.11% and 50.78% increase in data accuracy of two kind parameters, as compared to the data aggregation scheme.

Figure 5 shows the comparison of energy consumption between our scheme and two benchmark schemes in different transmission rounds. The results show that our scheme is more energy-efficient than the other two schemes. The energy consumption in the sensor-cloud environment is mainly due to the transmission from physical sensors to the gateway and sensing environmental data. The complex and energy-intensive computing tasks are processed in the cloud. In our scheme, only 11 physical sensors are activated to provide service, which achieves almost 75.59% decrease in energy consumption, as compared to the data aggregation scheme. As for the LEACH scheme, all the physical sensors are activated to provide services. The energy expenses are mainly due to the sensing, transmission, and computing of data. Furthermore, there are more repetitive message packets forwarded among the physical sensors. However, in our scheme, the communications among physical sensors are little. Because the data packets are sent to the gateway directly. Thus, a large amount of energy is conserved.

The energy variation directly reflects the network lifetime. In this experiment, both the LEACH and data aggregation schemes exploit 50 nodes. Figure 6 shows the comparison of network lifetime. In sum, this is about 1600 rounds in *p* = 0.1 scenario, 2100 rounds in *p* = 0.5 scenario, and 5800 rounds in data aggregation scenario. Obviously, our scheme has the longest lifetime.

## 6. Conclusions

Virtual sensor provisioning is one of the foremost tasks in sensor-cloud. The existing studies on the sensor cloud mainly consider the selection of all physical sensors, which results in a massive amount of energy consumption. Inspired by the machine learning methods, we propose a virtual sensor provisioning scheme to realize data similarity analysis. First, all physical sensors are divided into *l* classes by different changing trends. Then the *k*-means clustering algorithm is applied for each class. Finally, representative physical sensors are chosen to create a corresponding virtual sensors. In summary, we use fewer physical sensors to provide higher quality services. Meanwhile, our scheme reduces more energy consumption and prolongs the overall lifetime of the network. The experimental results show that our approach is efficient and suitable for virtual sensor provisioning tasks.

Nonetheless, there are still some issues that need further elaboration in future studies, such as virtual sensor provisioning under the heterogeneous environment, selection algorithm of the representative physical sensors.

## Figures and Tables

**Figure 1 sensors-20-01836-f001:**
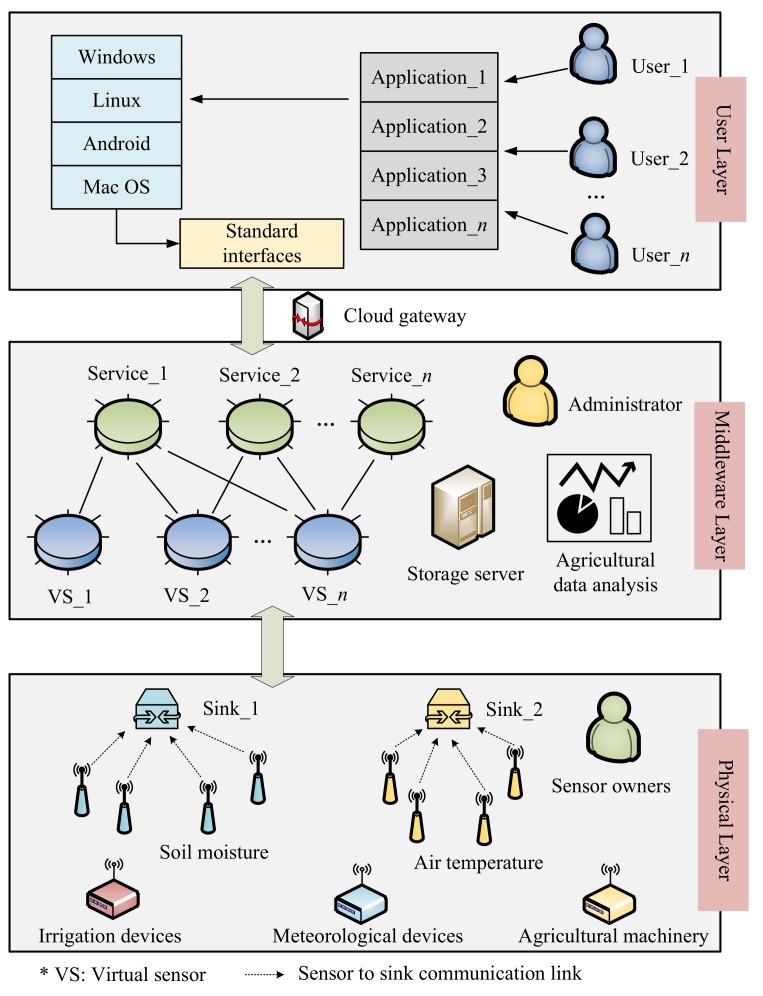
Layered structure of agriculture sensor-cloud.

**Figure 2 sensors-20-01836-f002:**
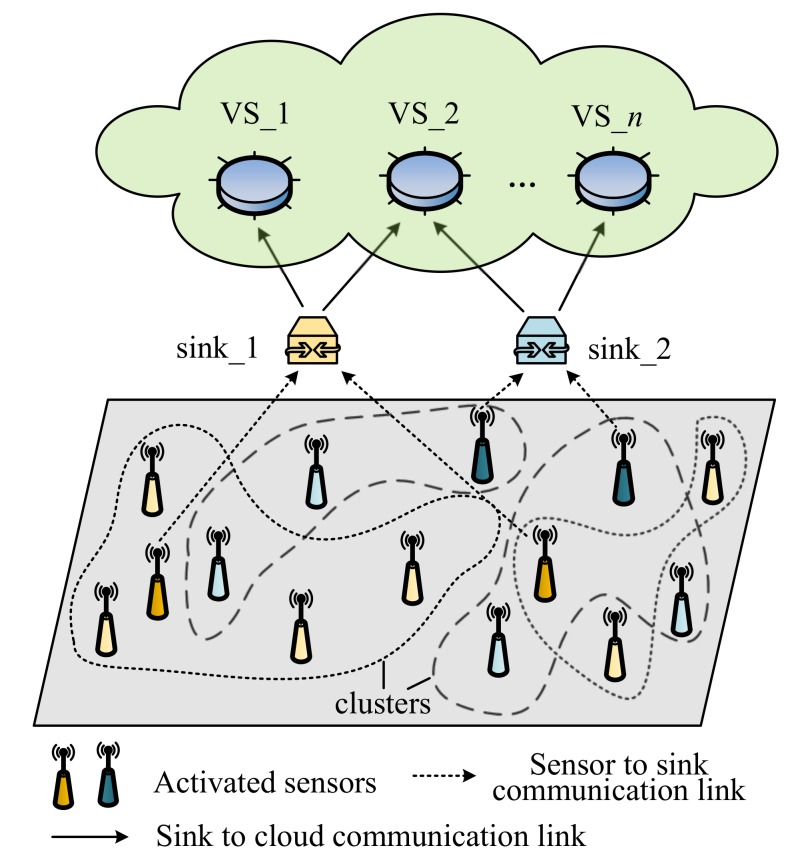
Virtualization of the physical sensors.

**Figure 3 sensors-20-01836-f003:**
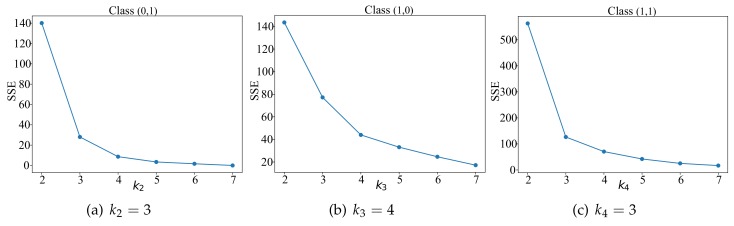
Select the optimal value of $k_{i}$.

**Figure 4 sensors-20-01836-f004:**
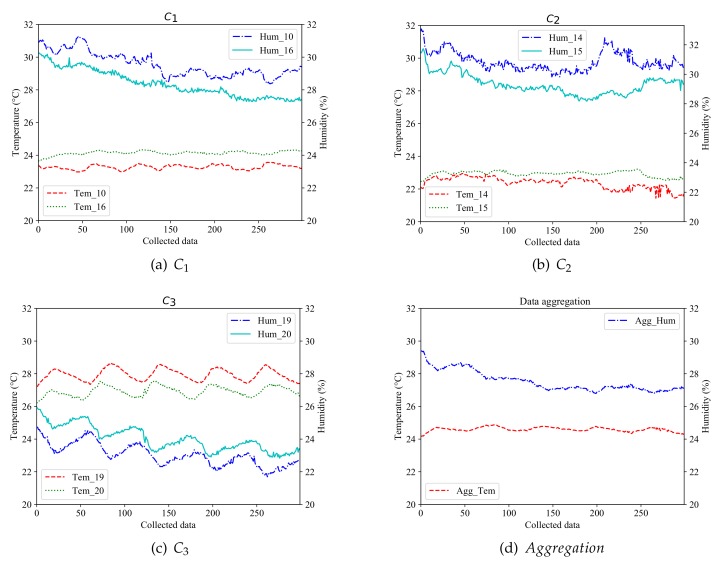
Comparative analysis for data accuracy.

**Figure 5 sensors-20-01836-f005:**
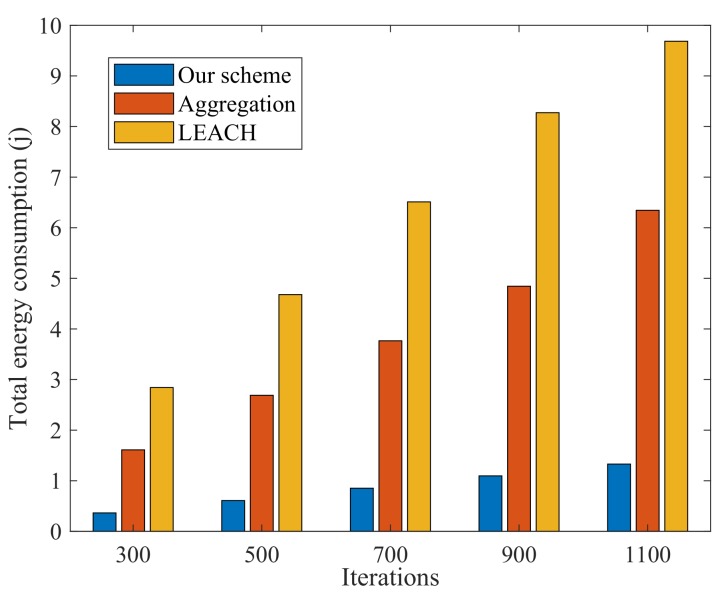
Comparative analysis for energy consumption.

**Figure 6 sensors-20-01836-f006:**
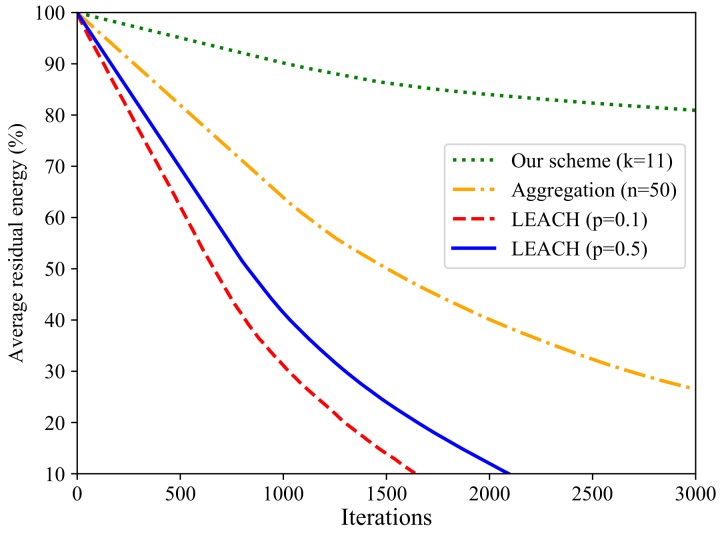
Comparative analysis for network lifetime.

**Table 1 sensors-20-01836-t001:** Simulation Parameters.

Parameter	Value
Simulation area	100 m * 100 m
Location of sink	(50 m, 100 m)
Number of nodes	50
Transmission range	100 m
Initial energy	0.3 j
Message length	256 bits

**Table 2 sensors-20-01836-t002:** Classification results.

Θ	Physical Sensors
(0,0)	{3,6,47}
(0,1)	{10,14,15,16,19,20}
(1,0)	{1,2,4,5,7,11,17,21,23,24,26,27,28,29,30
	31,32,33,34,35,36,37,38,39,48,49,50}
(1,1)	{8,9,12,13,18,22,25,40,41,42,43,44,45,46}

**Table 3 sensors-20-01836-t003:** Clustering results.

Θ	Cluster
(0,0)	{3,6,47}
(0,1)	{10,16}, {14,15}, {19,20}
(1,0)	{2,4,5,7,11,30,31,32,33,48,49,50}, {23}
	{1,17,29,35,36,38,39}, {21,24,26,27,28,34,37},
(1,1)	{8,9,12,13,40,41,42,43,44,45,}, {22,25}, {18,46}

**Table 4 sensors-20-01836-t004:** Data accuracy of our scheme.

Parameter	$\delta_{10}$	$\delta_{14}$	$\delta_{19}$	$\Delta_{\mathrm{os}}$
Temperature(T)	0.8797	0.6376	1.0537	0.8570
Humidity(H)	1.2364	1.6984	1.0439	1.3262

**Table 5 sensors-20-01836-t005:** Data accuracy of aggregation scheme.

Parameter	$\delta_{10}$	$\delta_{14}$	$\delta_{15}$	$\delta_{16}$	$\delta_{19}$	$\delta_{20}$	$\Delta_{\mathrm{agg}}$
Temperature(T)	1.3297	0.4875	2.2456	1.6793	3.3555	2.3367	1.9089
Humidity(H)	3.5117	4.5019	1.8327	3.3690	0.9170	2.0327	2.6942

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
