# Peer review of "Using Machine Learning Methods to Provision Virtual Sensors in Sensor-Cloud"

_sensors, 2020, doi:10.3390/s20071836_

Round 1
Reviewer 1 Report
+ The authors study the problem of energy-efficient virtual sensor provisioning based on machine learning methods.
+ The topic of the paper is interesting and under the scope of the journal. The analysis is rigorous, and the presented results provide insight information to the examined problem.
+ The adoption of machine learning approaches to deal with the virtual sensor provisioning problem is promising and interesting. However, machine learning acts as a block box when analytical solutions are not available, thus, the authors should justify some aspect in their analysis.
The authors should address the following comments to improve the quality of presentation of their manuscript and its technical depth.
- Initially, the usage of the English language should be improved. The reviewer proposes to the authors to have an expert in the usage of English language to check their manuscript.
- The authors should define the communication model in the sensors environment that they are studying that currently is totally missing. The authors should get information from a book, e.g., Singhal, Chetna, and Swades De, eds. Resource allocation in next-generation broadband wireless access networks. IGI Global, 2017, in order to better describe the communication model.
- The authors use the machine learning approach in order practically to achieve the classification. However, simpler classification algorithms have been proposed in the literature, e.g., Tsiropoulou, E., K. Koukas, and S. Papavassiliou. "A socio-physical and mobility-aware coalition formation mechanism in public safety networks." EAI Endorsed Trans. Future Internet 4 (2018): 154176, Lin, Chunhung Richard, and Mario Gerla. "Adaptive clustering for mobile wireless networks." IEEE Journal on Selected areas in Communications 15.7 (1997): 1265-1275. Those approaches are of much lower complexity compared to the proposed machine learning framework and the authors should discuss those alternatives in the related work.
- Following the previous comment, I believe that the machine learning framework may achieve more accurate classifications with the cost of increased computational complexity. The authors should include a section in the revised manuscript describing the computational complexity of the proposed framework, and more importantly, where the proposed framework will be implemented in a realistic implementation.
Reviewer 2 Report
This paper studied on how to achieve virtual sensor provisioning more efficiently. And authors present an energy-efficient virtual sensor provisioning scheme in which using a classical machine learning algorithm—k-means. The idea of their scheme is quite good and interesting.
Here are some points that need to be improved in this paper:
- The number mark formats of references are inconsistent;
- Please explain the meaning of ‘μ’ in Algorithm 1;
- What does Title 2 mean in Table 2? I suggest finding a clear title to replace ‘Title2’;
- The sub-heading of Figure 3 is not centered, it is recommended to readjust the layout of the figure;
- The number labels are duplicated in reference section;
- Some typo errors occurred in the manuscript, it is suggested to proof carefully;
- Authors are suggested to review more new and relevant research to support their research contribution. The following references should be considered:
[1]"Adaptive Data and Verified Message Disjoint Security Routing for Gathering Big Data in Energy Harvesting Networks," Journal of Parallel and Distributed Computing, vol. 135, pp. 140-155, 2020.
[2] "A Novel Load Balancing and Low Respon se Delay Framework for Edge-Cloud Network based on SDN" IEEE Internet of Things Journal. DoI: 10.1109/JIOT.2019.2951857, 2019.
[3] Machine Learning based Code Dissemination by Selection of Reliability Mobile Vehicles in 5G Networks. Computer Communications. vol. 152, 109-118, 2020.
Reviewer 3 Report
In this paper, the authors are proposing a virtual sesor provisioning scheme using sensor network readings with the middleware block between cloud and sensor devices. Traditional machine learning algorithms are applied on sensor reading classification and clustering purposes to reduce the energy consumption of the network compared to LEACH and aggregation algorithms. In general the paper has good theoretical study on application of machine learning paradigm to the considered problem of energy efficient data transmission of sensor readings. Medium writing and good presentation style in the paper. There are some parts which need to be rephrased or incomplete sentences as well.. However, it needs more detailed analysis in some sections. Therefore I recommend major revision.
- The performance improvements are not detailed. More numerical comparisons explanations and the reasons behind improvements compared to state of the art techniques mentioned (LEACH and aggregation algorithms) should be demonstrated.
- The referencing of references is not well aligned with the journal’s standard. For example Lemos 13, 14 should be Leamos [13,14] on page 2 line 53, etc for all the remaining as well.
- It is not clear to me if the classification algorithm is called liner regression? Logistic regression is instead used for classification purposes.
- Why are there 4 classes on Table 2. Is it because you are only considering temperature and humidity?
- Make the fonts of Fig. 3 larger
- Which reference is aggregation in Fig. 4?
- Give more explanations to fig. 4’s descriptions in terms of tradeoffs, numerical comparisons etc.
- Data accuracy discussion on page 12 is not clear. No numerical formula is given and it is not clear how the results in Fig. 6 d are not precise. Moreover, why you can choose 10, 14 and 19 to create virtual sernsor and services?
Minor comments
- Which paper is referred to on page 2 line 45 on “method in this paper the physical”?
- On page 2 line 52, “However numerous unnecessary”
- On page 2 line 53, “both costs of user and sensor”
- Sentences are incomplete or rephrased on page 3 line 67, 76 and on page 8 line 186
Round 2
Reviewer 1 Report
The authors have done an initial effort to improve the quality of presentation and the scientific depth of their manuscript. The authors should try to improve the communication model in the sensors environment that they are studying that currently is vague. I accept the fact that the authors have difficulties to find a book, and they may get advise from the following: Zhang, Yan, and Mohsen Guizani, eds. Game theory for wireless communications and networking. CRC press, 2011. Fragkos, G., "Disaster Management and Information Transmission Decision-Making in Public Safety Systems." 2019 IEEE Global Communications Conference. IEEE, 2019. Also, the authors should provide the derivations and some additional discussion regarding the complexity of the proposed framework. Some more discussion included in the corresponding section will improve the better understanding of the analysis by the audience.
Reviewer 3 Report
The authors have improved the paper based on my previous comments. My concerns have been addressed by authors.
Author Response
Thank you for your kindly instruction.
Round 3
Reviewer 1 Report
The authors have successfully addressed the reviewers' comments.